# Highly Stretchable and Sensitive Flexible Strain Sensor Based on Fe NWs/Graphene/PEDOT:PSS with a Porous Structure

**DOI:** 10.3390/ijms23168895

**Published:** 2022-08-10

**Authors:** Ping’an Yang, Sha Xiang, Rui Li, Haibo Ruan, Dachao Chen, Zhihao Zhou, Xin Huang, Zhongbang Liu

**Affiliations:** 1School of Automation, Chongqing University of Posts and Telecommunications, Chongqing 400065, China; 2Chongqing Key Laboratory of Materials Surface & Interface Science, Chongqing University of Arts and Sciences, Chongqing 402160, China

**Keywords:** porous structure, Fe NWs/Graphene/PEDOT:PSS, bridging, passivation and adhesion effect, three-dimensional conductive network

## Abstract

With the rapid development of wearable smart electronic products, high-performance wearable flexible strain sensors are urgently needed. In this paper, a flexible strain sensor device with Fe NWs/Graphene/PEDOT:PSS material added under a porous structure was designed and prepared. The effects of adding different sensing materials and a different number of dips with PEDOT:PSS on the device performance were investigated. The experiments show that the flexible strain sensor obtained by using Fe NWs, graphene, and PEDOT:PSS composite is dipped in polyurethane foam once and vacuum dried in turn with a local linearity of 98.8%, and the device was stable up to 3500 times at 80% strain. The high linearity and good stability are based on the three-dimensional network structure of polyurethane foam, combined with the excellent electrical conductivity of Fe NWs, the bridging and passivation effects of graphene, and the stabilization effect of PEDOT:PSS, which force the graphene-coated Fe NWs to adhere to the porous skeleton under the action of PEDOT:PSS to form a stable three-dimensional conductive network. Flexible strain sensor devices can be applied to smart robots and other fields and show broad application prospects in intelligent wearable devices.

## 1. Introduction

Flexible sensors have shown extensive application prospects in the fields of health monitoring, as well as artificial intelligence, because of their excellent flexibility, good biocompatibility, small size, and light weight [1,2]. In order to apply flexible sensors in these fields, the fabricated sensors need to meet the requirements of mechanical flexibility, linearity, sensitivity, and stability [3,4]. There are many variations of classification methods for flexible sensors, and the common classification is based on the type of measurement, including biosensors, chemical sensors, and physical sensors [5,6,7]. Biological sensors mainly detect the sensitive properties of biological substances; chemical sensors are commonly used to detect chemical substances, while physical sensors are measured by detecting physical quantities such as resistance, capacitance, and voltage. Physical sensors can be divided into capacitive [8,9], piezoelectric [10], and triboelectric [11] according to their detection principles. Among them, piezoresistive sensors have attracted extensive research interest because of their simple structure, simple preparation process, low energy consumption, and high sensitivity [12,13].

One of the main factors affecting the performance of resistive flexible strain sensors is the selection of sensitive element materials. Common strain materials include metal series materials [14,15,16], carbon series materials [17,18,19,20], and other materials [21]. Considering that metal series materials have extremely high electrical conductivity and are easy to form conductive networks [22], the prepared flexible sensor has a high sensitivity factor and is therefore commonly used to construct sensitive units of sensors. Metal nanowires, due to their large aspect ratio, lead to the existence of relatively good flexibility, and a large number of metal nanowires can easily form a misaligned conductive network, which allows them to be used as sensing functional materials for flexible strain sensors [23,24,25]. At present, a lot of research is focused on gold and silver nanowires, which exhibit excellent sensing properties due to their stable chemical properties; however, the preparation process of the complex, expensive raw materials and the restricted wide strain range due to higher aspect ratios and weak adhesion on flexible substrates have limited their development to some extent [26], and despite many efforts to improve the wide strain range [27], they still suffer from the disadvantage of poor stability performance over a wide strain range. Fe NWs (Iron nanowires) are often used in electromagnetic shielding due to their superior shielding function [16], and considering that they are metal nanowires, they have certain sensing functions, but the current sensing performance is poor. How to improve the sensing performance of Fe nanowires has become the direction of research [28].

The sensor performance of Fe NWs is poor because they are highly susceptible to oxidation by air. In order to slow down the process of oxidation, a common method is to coat Fe NWs, considering graphene sensors exhibit high-sensitivity sensing performance due to their excellent mechanical and electrical properties; however, they have the disadvantage of a narrow strain range [29,30]. Therefore, the innovative combination of Fe NWs and graphene, using the good electrical conductivity of Fe NWs to act as a bridge for the graphene, can improve the narrow strain range of graphene. The way of using graphene as a passivation layer for Fe NWs to slow down the oxidation process can improve the problems such as the susceptibility of Fe NWs to sense failure. When constructing Fe NWs/Graphene sensitive units, the sensitive unit structure needs to be designed because Fe NWs tend to agglomerate, resulting in the inhomogeneous dispersion of Fe NWs and graphene. Common sensing structures include the surface structure [26], internal structure [31], fabric structure [4], and bionic structure [32,33]. Surface structures and bionic structures can improve the sensing performance of piezoresistive flexible strain sensors well, but they are not effective for tensile sensing performance and also have the problem of poor stability [34], and fabric structures can better combine sensitive materials and the human body but lack the defects of elasticity which form a low strain range [35]. Surface and fabric structures cannot improve the agglomeration properties of Fe NWs, while the porous skeleton present in the internal structure can not only improve the agglomeration phenomenon but also enhance the sensing performance within a certain range [31,36]. The most commonly used method for constructing porous structures is the template sacrifice method [37,38,39], but the formed pore structure is limited by many factors, and the preparation process is complicated, while the well-synthesized polyurethane foam can form a uniform pore structure and its pore size can be self-selected, which can easily form a uniform conductive network and improve the sensing performance of Fe NWs and graphene, while the substrate material for flexible sensors is usually chosen as PDMS (Polydimethylsiloxane), and the polyurethane foam and PDMS have a small difference in elastic modulus and cause a small hysteresis. Thus, the polyurethane foam structure is introduced as the substrate of the sensor-sensitive unit [40,41,42].

The sensor composed of Fe NWs/Graphene tends to detach the sensitive material from the polyurethane foam skeleton after multiple uses, resulting in sensing performance becoming worse. Considering the introduction of sticky or colloidal polymers mixed with the use of [43], PEDOT:PSS (poly (3,4-ethylenedioxythiophene): poly (styrene sulfonate)) is a conductive polymer material with its own viscosity. Ron E et al. [44] prepared a flexible strain sensor that can maintain 1000 stable cycles at 20–30%, which can force Fe NWs and graphene to adhere well to polyurethane foam and improve the drawback of poor stability of Fe NWs/Graphene, providing the possibility of preparing flexible sensors with high sensitivity, wide strain range, and high stability.

Therefore, a flexible strain sensor of Fe NWs/Graphene/PEDOT:PSS under a porous structure was prepared in this study. Based on the porous structure and cooperative conductive network, the brittle damage effect of Fe NWs located on the cytoskeleton, the passivating effect of graphene and the linking effect on fracture, and the stable PEDOT:PSS during stretching constitute a flexible strain sensor. The test results show that the sensor is composed of three strain materials, Fe NWs/Graphene/PEDOT:PSS, and the sensor prepared by soaking PEDOT:PSS for one time has better performance, with high sensitivity (10.65), wide strain range (0%~100%), local high linearity (98.8%), good stability, and so on. In addition, the flexible strain sensor based on the composite material can detect the motion of human fingers and elbows and can also detect grasped objects with different diameters, showing potential application prospects in flexible wearable electronic devices.

## 2. Experiment

### 2.1. Materials

A PDMS elastomer kit (Sylgard 184) was purchased from Dow Coring. Anhydrous ethanol (C_2_H_6_O, AR ≥ 99.7%) was purchased from Chengdu Kelon Co., Ltd. (Chengdu, China). Deionized water (18.25 MΩ) that was used in this experiment was obtained from an ultrapure water system (GYJ2-20L-s, Chongqing Huachuang). PEDOT:PSS aqueous solution was purchased from Hunan Nacheng Printing Electronic Technology Co., Ltd. (Hunan, China). Polyurethane sponge was purchased from Jiangsu Xingyike New Material Co., Ltd. (Jiangsu, China).

### 2.2. Preparation of Fe NWs/Graphene/PEDOT:PSS Strain Sensor

The manufacturing process of Fe NWs/Graphene/PEDOT:PSS is shown in Figure 1a. First, polyurethane foam is cut into 3 cm × 1 cm size using deionized water and ethanol washed three times, then immersed in Fe NWs solution, graphene solution with a mass fraction of 0.5%, and PEDOT:PSS solution with a solid content of 5%, and then vacuum drying was carried out. In the process of soaking, the soaking time was 15 min, and after the immersion was completed, the sensitive layer was dried in a vacuum drying box (60 °C) (The operation of soaking PEDOT:PSS one more time is after soaking Fe NWs, graphene, and PEDOT:PSS solution in sequence). Then, mix well component A and component B of PDMS according to 10:1, put them into a vacuum drying oven to vacuum and remove bubbles, and place the PET film on the homogenizer (the rotating speed is set to 500 r/min and the homogenizing time is 15 s). Cast the PDMS with bubbles removed on the PET (Polyethylene terephthalate) film, prepare two pieces of PDMS films with a thickness of 1 mm, and dry them in a 60 °C blast drying oven for 15 min, turning PDMS into a semi-solidified state; the schematic diagram is shown in Figure 1b. Finally, the copper foil electrode was pasted onto the fixed position of the semi-solidified PDMS film. The prepared sensitive unit was placed on the PDMS which was pasted with the copper foil electrode, bonded with another semi-solidified PDMS film, and dried in the drying oven (3 h) to prepare the Fe NWs/Graphene/PEDOT:PSS flexible strain sensor; the physical drawing is shown in Figure 1c.

### 2.3. Characterization

The microscopic morphology of Fe NWs, Fe NWs/Graphene, Fe NWs/PEDOT:PSS, and Fe NWs/Graphene/PEDOT:PSS were observed by field emission scanning electron microscopy (SEM, Thermo Scientific Apreo2C), as shown in Figure 2a–d, to analyze the relationship between the sensing performance and the microscopic morphology of Fe NWs/Graphene/PEDOT:PSS flexible strain sensors. The film tensile test system (XN60-502N) was used to carry out the cyclic tensile test; two ends of the sensor were fixed with clamps, and the original sensor length was 20 mm. The relationship between the relative resistance and the displacement was collected and analyzed by computer: the strain range was 0–100%, the rate was multiplied changed from 1 mm/s–4 mm/s, and all sensing performances were evaluated.

## 3. Results and Discussion

### 3.1. Microcosmic Structure

SEM photographs show that Fe NWs are composed of multiple interlaced iron nuclei with a bead-chain structure, and bead-chain Fe NWs contact each other to form a conductive network, as shown in Figure 2a. The lamellar structure of graphene covers Fe NWs, which slows down the oxidation process of Fe NWs and enhances the electrical conductivity of graphene, and the fibrous PEDOT:PSS forces the Fe NWs to firmly adhere to the polyurethane foam structure, as shown in Figure 2b,c. The Fe NWs are tightly attached to the pore surface and tightly wrapped by graphene, and PEDOT:PSS is firmly adhered to both in the polyurethane foam structure, so that they can form a property structure, so that it can form a flexible resistive sensor with stable performance, as in Figure 2d.

### 3.2. Fe NWs/Graphene/PEDOT:PSS Strain Sensor

The strain characteristics of flexible strain sensors can be tuned by adjusting the composition of the sensor-sensitive layer. Therefore, we designed and prepared different sensitive layers and tested the related properties. Figure 3a–c shows pictures of different tensile states corresponding to 0%, 80%, and 100% stretching, respectively, and after repeated tests, the strain range of this flexible sensor was verified to be 0–100%. Figure 3d–g shows the tensile properties. Here, the change of the relative resistance is defined as ΔR/R = (R_1_ − R)/R, where R is the initial state resistance, R_1_ is the resistance under the current tensile strain, and ΔR is the change in resistance. Four different experiments were designed in order to analyze which materials were selected to prepare the flexible strain sensors. Firstly, the linearity and sensitivity of Fe NWs are explored. It is found that the linearity of Fe NWs is poor because the average aspect ratio of Fe NWs is 350, and during the stretching process, due to the limitation of aspect ratio, Fe nanowires are brittle and easy to fracture with a large specific surface area. It is easy to produce a large contact area with the surrounding environment, leading to the intensification of the oxidation process and worsening the linearity. When the strain exceeds 50%, the sensing performance fails due to the lack of connection; the Fe NWs’ connection point breaks under high tension. In order to slow down the oxidation process of Fe NWs and improve the linearity of Fe NWs, graphene or PEDOT:PSS were considered, and the linearity was improved (R^2^ = 96.4% and R^2^ = 97.6%). Thus, the simultaneous addition of graphene and PEDOT:PSS to Fe NWs was considered. When the stress is 0–10%, the linearity is poor; when the strain is 10–100%, the linearity is good (R^2^ = 98.8%). This is because, at a smaller strain, the Fe NWs are surrounded by more graphene and PEDOT:PSS, resulting in a certain degree of resistance on the sensitive layer that would have produced cracking. At this point, the resistance is greater than the stress, leading to the weakening of its response. Therefore, the linearity is poor, and when the strain is more than 10%, the stress is higher, and the resistance generated by more graphene and PEDOT:PSS has less effect on the stress and has a negligible effect on the linearity, so the linearity is better.

The gauge factor (GF) is one of the essential elements in a strain sensor to evaluate its performance. GF is defined as (ΔR/R)/ε, where ΔR/R is the relative change in resistance of the strain sensor and ε is the applied strain. Generally speaking, the flexible sensor composed of Fe NWs/Graphene/PEDOT:PSS has the highest sensitivity, followed by Fe NWs, while Fe NWs/Graphene and Fe NWs/PEDOT:PSS have the lowest sensitivity, as shown in Figure 3d–g. The sensitivity of Fe NWs/Graphene/PEDOT:PSS is about 10.65. With the increase in tensile stress, microcracks appear in the sensitive layer, the crack gap, and the number of cracks increases, leading to a great change in the relative resistance and a high sensitivity of the flexible sensor. However, in the small strain range (0–10%), the sensitivity coefficient is very low. Too much graphene and PEDOT:PSS attached to the Fe NWs will hinder the increase in the crack gap at small tensile strains, and when the number of cracks is also less, it results in the slow growth of resistance and low sensitivity of the flexible sensor.

In order to verify whether PEDOT:PSS can improve the stability of the flexible sensors, repeated performance tests of the flexible were performed on the flexible sensor. Fe NWs were not tested for repeatability because they failed to sense at large strain ranges. The others were tested for 3500 repetitions at 80% strain, and it was found that the stability of Fe NWs/Graphene flexible strain sensors was poorer. After 1000 repetitions of stretching, a large amount of sensitive material fell off from the substrate and the rate of change of resistance was dramatically different than before, as shown in Figure 4a. Due to the presence of PEDOT:PSS, the stability of the Fe NWs/PEDOT:PSS sensor can be improved to a certain extent, but the sensitivity is poor, as shown in Figure 4b. By combining all three, the Fe NWs/Graphene/PEDOT:PSS flexible strain sensor maintains good stability while maintaining high sensitivity, as shown in Figure 4c.

It is also necessary to investigate whether the different number of impregnations of PEDOT:PSS has an effect on the sensing performance. Considering the long time period of impregnation, the number of immersions should not be too many in order to avoid the exposed Fe NWs being exposed to air for a long time during the immersion transfer process, which leads to the oxidation of the Fe NWs and deterioration of the performance. Therefore, two samples with different immersion times were designed, sample 1 for PEDOT:PSS immersion once, and sample 2 for immersion twice. Their linearity and sensitivity were tested, respectively. The performance diagram of sample 1 is shown in Figure 3g, and the performance diagram of sample 2 is shown in Figure A1. It is found that the sensitivity and linearity of the flexible sensor have a certain degree of decline after immersion two times compared with one. This is because, after dipping twice, part of the Fe NWs fell off from the polyurethane foam, and the content of Fe NWs, which plays the main role of conducting, reduced relatively and linearity, worsening the sensitivity. Therefore, in the performance test of the Fe NWs/Graphene/PEDOT:PSS flexible strain sensor, a sample soaked in PEDOT:PSS once was selected as the test sample.

Whether the stability performance will deteriorate with time is also the focus of the study. Time interval tests were conducted on the prepared sensors to test the stability of the sensors after 15 and 30 days, as shown in Figure 5. After 15 days, the repeatability is basically the same as before, although the performance of the sensor fluctuated to some extent after 30 days, the overall trend remained unchanged and the stability was good, indicating that the introduction of PEDOT:PSS could strengthen the adhesion of Fe NWs and graphene adhesion with polyurethane foam and improve their reproducibility.

In addition to investigating the linearity, sensitivity, and thermal stability performance of the sensor, other performance tests were performed. To determine whether the sensing performance would be reduced or even lost under small strain, we tested small tensile stress and observed the relationship between its relative resistance and displacement. The results show that the relative resistance increases by 2.06 × 10^−3^ when the strain is changed from 0% to 1% and the relative displacement is changed from 0 to 2 mm (Figure 6a). By comparing the variation law of the relative resistance of the flexible sensor under large tensile strain, it is found that when the strain changed by 1%. Its relative resistance changed by about 3 × 10^−3^, which is less different from the change of relative resistance under small strain, indicating that the sensor is not only suitable for large tensile strain but is also suitable for small strain detection.

The matrix used to fabricate flexible sensors is polymer PDMS, which has common problems such as response delay and cyclic instability [45]. The response time and stability of the flexible sensor can be improved by changing the material and structure of the sensitive layer. The response time of the flexible sensor composed of carbon material is about 170 ms, while the response time of the above flexible sensor is 260 ms (Figure 6b). Although the response time of the flexible sensor is longer than that of carbon material, its wide strain range can achieve a balance between strain range and response time. For the sensor prepared by polymer composites, the hysteresis is due to the fact that the Young’s modulus of the matrix material and the conductive material cannot be well adapted, resulting in different strain states during loading and release and with the existence of hysteresis, which is partially eliminated by pre-stretching (Figure 6c), but under large tensile stress, there is a lag gap between the elastic modulus changes of polyurethane foam and the substrate, resulting in a certain degree of hysteresis still existing, which can be improved by changing the structure of the flexible sensor. To analyze the effect of tensile frequency on stability, the stability was tested at different rates (1 mm/s, 2 mm/s, 4 mm/s) (Figure 6d), and the sensor has good stability regardless of tensile rate, so the combination of Fe NWs, carbon material, and PEDOT:PSS could not only increase the strain range, but also improve its stability and provide support for its application in intelligent robots, smart medical treatment, and other fields.

### 3.3. Analysis of Sensing Mechanism of Fe NWs/Graphene/PEDOT:PSS Flexible Strain Sensor

The sensing mechanism of the resistive flexible strain sensor mainly includes the geometric effect, conductive phase separation, tunneling effect, crack propagation, and percolation theory [46,47,48,49,50]. The sensor in this paper can show high sensitivity and the linear response over a large strain range, which can be explained by the interaction of Fe NWs, graphene, and PEDOT:PSS compounded under the sponge structure matrix.

The current state of research shows that when a flexible strain sensor is subjected to stress, the first and most significant sensing mechanism is the geometric effect. When the whole sensitive layer is considered as a resistive device, the variation of its resistance is determined by R =$\;\rho \frac{l}{s}$, where ρ is the resistivity of the sensitive unit, *l* is the length of the sensitive unit, and *s* is the cross-sectional area of the sensitive unit. When the flexible sensor is subjected to an external force, the length *l* of the sensitive body increases, and the cross-sectional area s becomes smaller due to the high elasticity of the sensor itself, resulting in an increase in the resistance of the entire flexible strain sensor. In the application of stress, when the stress disappears, the length and cross-sectional area of the sensor return to the initial state, and the resistance value also becomes the original, which depends on the magnitude of the stress. Since the Fe NWs with the linear structure are distributed in the porous structure, the linear distribution of Fe NWs is curved when no force is applied, the linear distribution of Fe NWs is linear under stress, and the linear network can be formed under large strain, meaning the sensor shows a linear response over a large strain range.

The second is the crack propagation mechanism. The porous three-dimensional structure selected in this paper is conducive to the formation of an effective conductive network. The excellent mechanical properties and high aspect ratio of Fe NWs constitute the main conducting pathway, but there are defects of uneven distribution, few conducting paths, and easy oxidation, which cannot form a stable conductive network. Graphene can be used as the passivation layer for Fe NWs while maintaining good conductivity, and, combined with the adhesion property of PEDOT:PSS, the prepared Fe NWs/Graphene/PEDOT:PSS flexible strain sensor can form a uniform and steady conductive network. Under the effect of strain, the low modulus of the elasticity of the flexible substrate leads to the easy generation of tensile deformation, and under the action of an interfacial bonding force, the sensitive element is very easy to form cracks and expand in all directions, and the existence of such cracks leads to the difficult transmission of electrons and affects the magnitude of resistance, but due to the existence of Fe NWs in it, it leads to the conduction of the region that would have produced cracks and further improves the stability of the conductive network, enabling the Fe NWs/Graphene/PEDOT:PSS flexible strain sensor to maintain a linear response under large strain. The schematic diagram of the mechanism analysis is shown in Figure 7.

Figure 8 shows the performance of the sensor composed of metal nanofillers, carbon materials, and other functional materials. By comparison, it is found that some sensors can achieve larger strain but have poor repeatability, while others are more sensitive. Although the sensor prepared in this paper has some defects in response time and its strain range is limited, it has good performance in repeatability, strain range, linearity, and sensitivity, which can better meet the needs of wearable electronic devices.

### 3.4. Application of Fe NWs/Graphene/PEDOT:PSS Flexible Strain Sensor

Based on the excellent performance of the Fe NWs/Graphene/PEDOT:PSS flexible strain sensor, it can be used as a skin-attached, wearable device for monitoring physiological signals and the large-scale movements of the human body. For example, strain sensors can detect the motion state of the human body by connecting the sensor to fingers, elbows, etc. The rate of change of resistance varies with the bending angle of each part (when the bending angle varies from 0 to 45° and 90°). Generally speaking, the larger the strain angle, the higher the rate of change of resistance, as shown in Figure 9a,b, where the flexible strain sensor is located on the finger, elbow, or other parts. When the strain angle is the same, the rate of change of resistance at different parts is basically the same, which proves that the flexible sensor has good repeatability. In addition, the flexible strain sensor can also be used to judge objects with different diameters, the flexible sensor shows different responses to grasp force signals, which can be used to judge the shape of grasping objects, as shown in Figure 9c,d. Therefore, the skin-mounted Fe NWs/Graphene/PEDOT:PSS strain sensor has great potential for monitoring human physiological signals and body movements. It provides important information for the feedback control of robotic systems and prostheses, which can be widely used in intelligent robots, smart medical treatment, and other fields.

## 4. Conclusions

In summary, we have successfully developed a flexible strain sensor based on Fe NWs/Graphene/PEDOT:PSS. The sensitive layer was prepared by combining different materials by immersion and varying the soaking time of PEDOT:PSS. A substrate with a thickness of 1 mm was prepared by a rotating coater, and then the substrate and the sensitive layer were bonded together to prepare a flexible strain sensor. It was found that the sensor with Fe NWs/Graphene/PEDOT:PSS hybrid and PEDOT:PSS immersion once in a polyurethane foam substrate exhibited high sensitivity (GF = 10.65) and reliable repetitive performance (3500 times) under the combined effect of geometric effects and a crack extension mechanism. Based on these excellent sensing properties of the prepared sensor, we demonstrate its practical applications in detecting various human signals and large-scale motions, including finger and elbow movements, and their perception at different diameters, showing its potential as a wearable electronic device for intelligent robots and health monitoring, but the weak strain and multimodal sensor needs to be studied further.

## Figures and Tables

**Figure 1 ijms-23-08895-f001:**
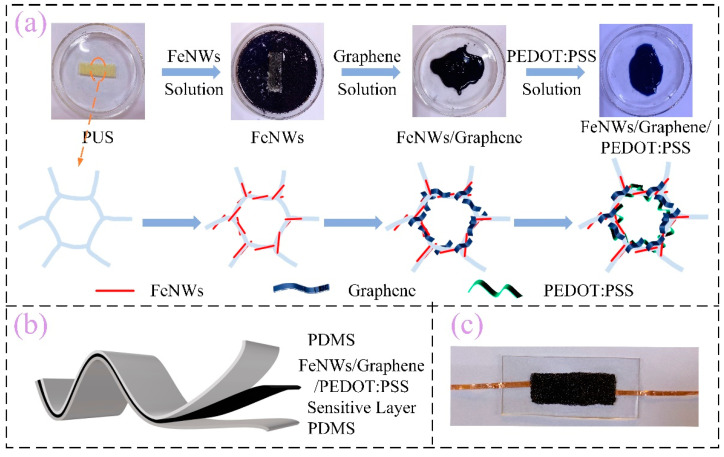
Fe NWs/Graphene/PEDOT:PSS sensor. (**a**) Preparation flow chart, (**b**) Schematic diagram, (**c**) Physical picture.

**Figure 2 ijms-23-08895-f002:**
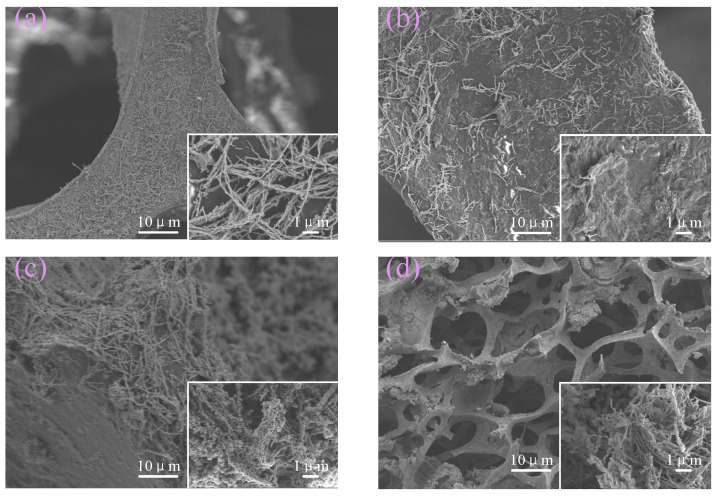
SEM images of (**a**) Fe NWs, (**b**) Fe NWs/Graphene, (**c**) Fe NWs/PEDOT:PSS. (**d**) Fe NWs/Graphene/PEDOT:PSS.

**Figure 3 ijms-23-08895-f003:**
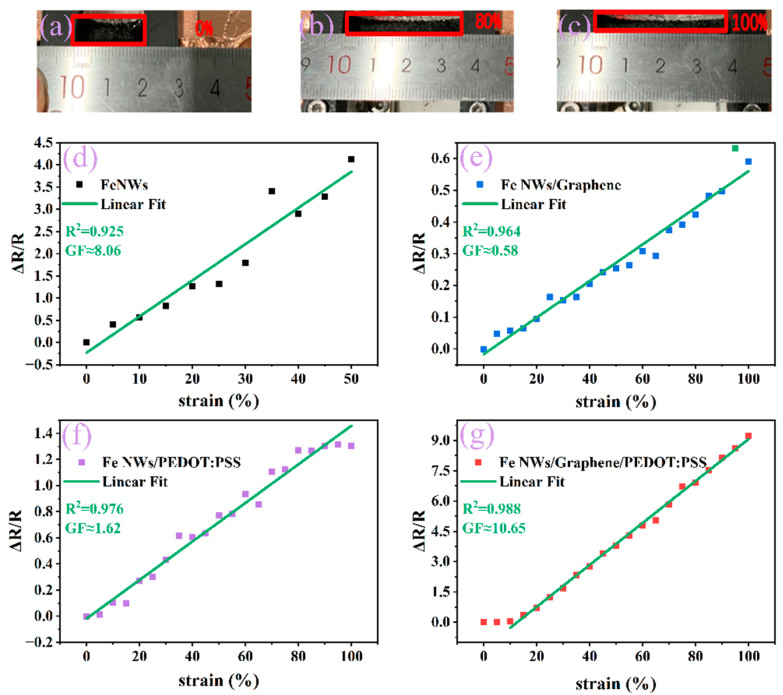
(**a**–**c**) Photographs of a strain sensor belt clipped on two clamps before and after stretching up to 80% and 100%, respectively. Linearity and sensitivity of (**d**) Fe NWs, (**e**) Fe NWs/Graphene, (**f**) Fe NWs/PEDOT:PSS, (**g**) Fe NWs/Graphene/PEDOT:PSS.

**Figure 4 ijms-23-08895-f004:**
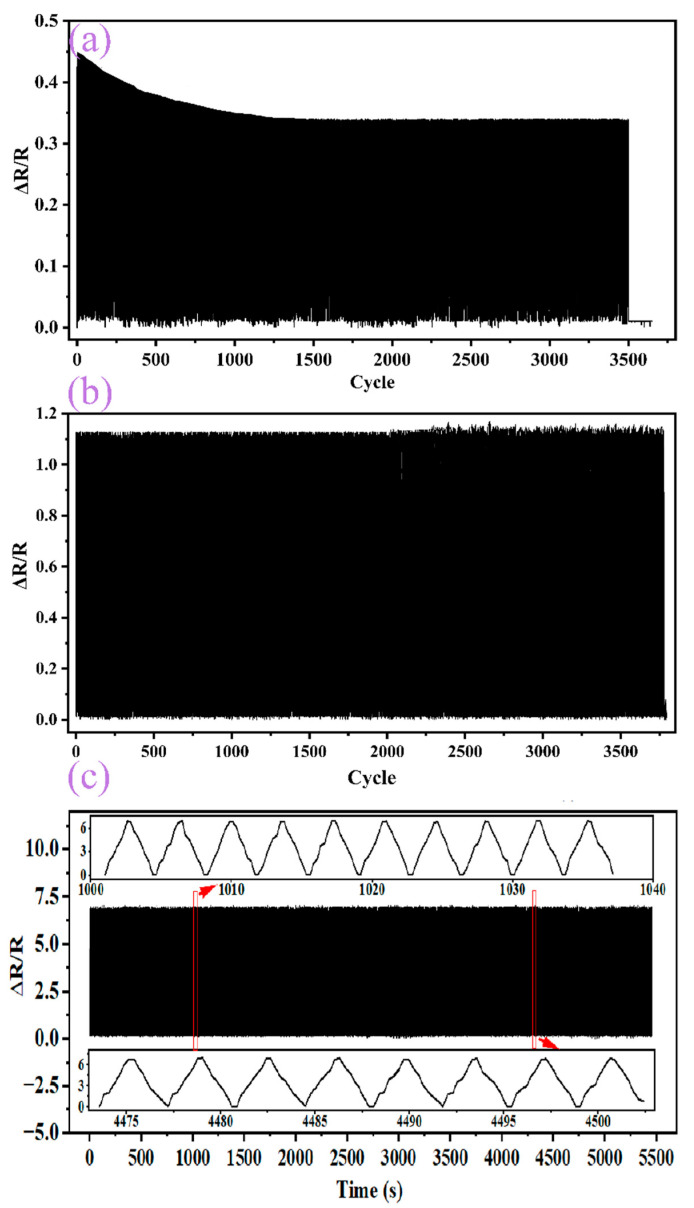
Repeatability of (**a**) Fe NWs/Graphene flexible strain sensor, (**b**) Fe NWs/PEDOT:PSS flexible strain sensor, (**c**) Fe NWs/Graphene/PEDOT:PSS flexible strain sensor under 80% strain.

**Figure 5 ijms-23-08895-f005:**
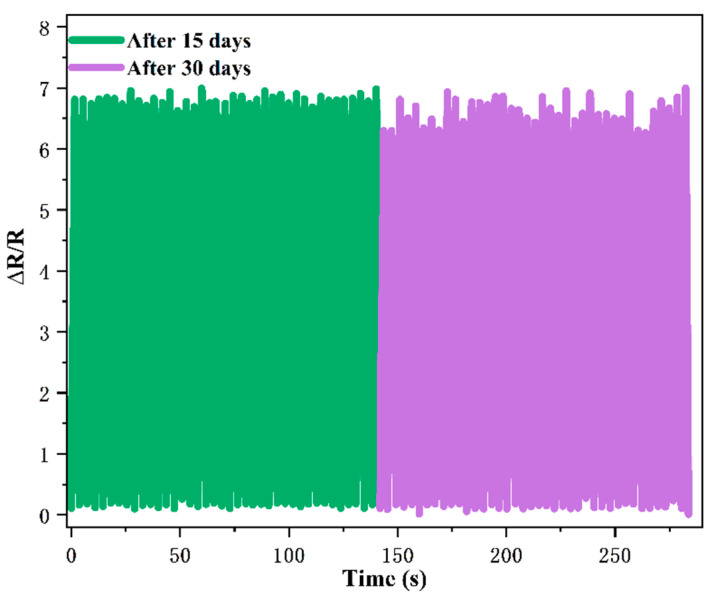
Comparison of repeatability of Fe NWs/Graphene/PEDOT:PSS strain sensor after 15 days and 30 days.

**Figure 6 ijms-23-08895-f006:**
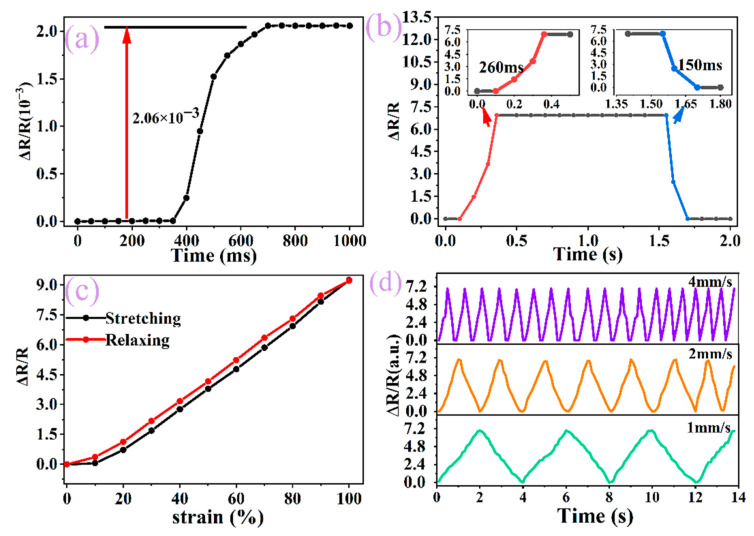
(**a**) Relative resistance-strain curve of Fe NWs/Graphene/PEDOT:PSS strain sensor under 1% strain, (**b**) Response time for loading and unloading at 80% strain, (**c**) Hysteresis curve with strain from 0% to 100%, (**d**) Relative resistance-strain curve of strain sensor at two times rate.

**Figure 7 ijms-23-08895-f007:**
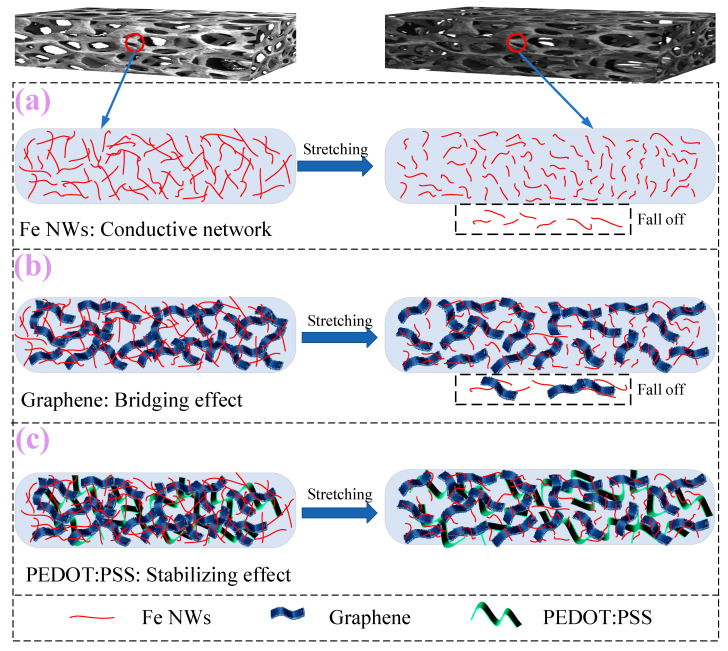
Analysis of sensing mechanism.

**Figure 8 ijms-23-08895-f008:**
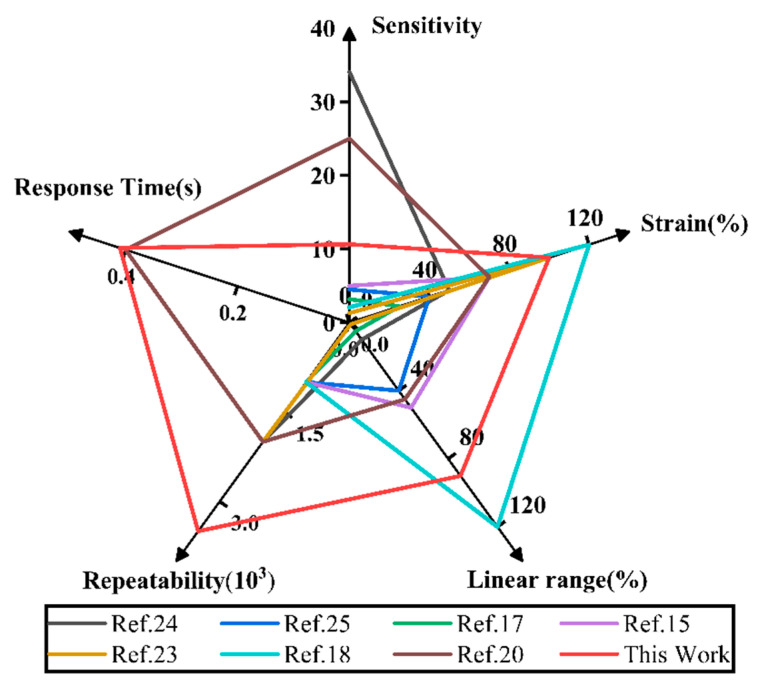
Performance comparison charts.

**Figure 9 ijms-23-08895-f009:**
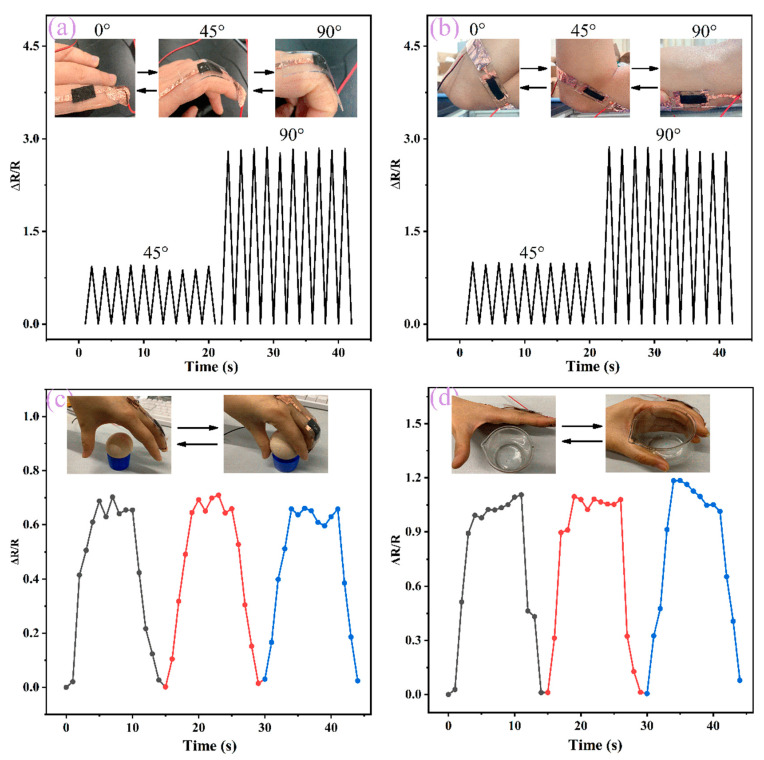
(**a**,**b**) Response curves of fingers and elbows at different bending angles. (**c**,**d**) Response curve when grasping objects with different diameters.

## Data Availability

Not applicable.

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
