# Peer review of "Highly Stretchable and Sensitive Flexible Strain Sensor Based on Fe NWs/Graphene/PEDOT:PSS with a Porous Structure"

_ijms, 2022, doi:10.3390/ijms23168895_

Round 1
Reviewer 1 Report
This manuscript by Yang et. al. provided a flexible strain sensor device with Fe NWs/Graphene/PEDOT: PSS materials added under a porous structure were designed and prepared, and the effects of adding different sensing materials and the different number of dips with PEDOT: PSS on the device performance were investigated. Overall, the whole manuscript was well-organized, and the information provided in this study and the experimental methodology are interesting. However, the authors could have explained this manuscript more thoroughly, I highly recommend its publication after a minor revision with the following comments addressed.
1. Introduction
Are there other methods that should be discussed in the introduction as well, some chem/biosensors can also be discussed as well (Journal of Materials Chemistry B 7 (16), 2613-2618; Microchimica Acta 186 (5), 1-9; Analyst 145 (10), 3598-3604)
2. In the Figure 3g and Figure S1, Why the R/R is Zero in less than 10% strain
3. The conclusion looks fine, and the main limitation also should be discussed as well.
4. There are a few areas where the English could be improved, such as some past and present tense.
5. There are some grammatical errors in this manuscript such as continuously forgetting to add ‘a’ or ‘the’ before a specific word which limits the clarity of the author’s writing.
Author Response
Reply to reviewer #1
Reviewer 2 Report
Manuscript titled „Highly Stretchable and Sensitive Flexible Strain Sensor Based on Fe NWs/Graphene/PEDOT:PSS With a Porous Structure” is focused about the material and its properties for sensorics application. It is written very well, the results are presented clearly and logically. In my opinion, it should be accepted for publication after some very small changes.
1. First of all, all the abbreviations should be explained – NWs, PEDOT, PSS, PET… I think that Authors assumed this is generally known knowledge, however not every reader can remember exactly what is polyethylenodioxythiophene, if is not similar with polymers. The figure of structure would also be beneficial.
2. „Polyurethane sponge bought from commercially” – what does it mean? From what company?
3. In my opinion the introduction part is should be re-written in places where authors use i.e. „and so on…” (first sentence) – in introduction are too many colloquialisms.
The rest of manuscript does not raise any objections.
Author Response
Reply to reviewer #2
